# Beta-Lactam vs. Fluoroquinolone Monotherapy for *Pseudomonas aeruginosa* Infection: A Systematic Review and Meta-Analysis

**DOI:** 10.3390/antibiotics10121483

**Published:** 2021-12-03

**Authors:** Eric Reid, Ryan W. Walters, Christopher J. Destache

**Affiliations:** 1School of Medicine, Creighton University, Omaha, NE 68178, USA; RyanWalters@creighton.edu (R.W.W.); chrisdestache@creighton.edu (C.J.D.); 2School of Pharmacy & Health Professions, Creighton University, Omaha, NE 68178, USA

**Keywords:** fluoroquinolone, beta-lactam, *Pseudomonas aeruginosa* infection, systematic review

## Abstract

Introduction: *Pseudomonas aeruginosa* (PA) is a leading cause of healthcare-associated infections. A variety of antibiotic classes are used in the treatment of PA infections, including beta-lactams (BLs) and fluoroquinolones (FQs), given either together in combination therapy or alone in monotherapy. A systematic review and meta-analysis were performed to evaluate the therapeutic efficacy of BL agents versus FQ agents as active, definitive monotherapy in PA infections in adults. Methods: Comprehensive literature searches of the Medline and Scopus electronic databases, alongside hand searches of the Cochrane Database of Systematic Reviews, PubMed, and Google Scholar, were performed without a time restriction to identify studies published in English comparing BL and FQ agents given as monotherapy for PA infection in hospitalized adults for which mortality, bacteriological eradication, or clinical response was evaluated. One reviewer screened search results based on pre-defined selection criteria. Two reviewers independently assessed included studies for methodological quality using NIH assessment tools. Two fixed-effects meta-analyses were performed. Results: A total of 368 articles were screened, and six studies involving 338 total patients were included in the meta-analysis. Upon evaluation of methodological quality, two studies were rated good, three fair, and one poor. A meta-analysis of three studies demonstrates FQ monotherapy is associated with significantly improved survival compared to BL monotherapy for patients with PA bacteremia (OR, 3.65; 95% CI, 1.27–10.44; *p* = 0.02). A meta-analysis of three studies demonstrates FQ monotherapy is associated with equivalent bacteriological eradication compared to BL monotherapy for PA pneumonia or skin and soft tissue infection (RD, 0.07; 95% CI, −0.09 to 0.24; *p* = 0.39). Conclusion: The meta-analyses demonstrate FQ monotherapy significantly improves survival in PA bacteremia and is associated with similar rates of bacteriological eradication in pneumonia and skin and soft tissue infection caused by PA compared to BL monotherapy. However, more research is needed to make meaningful clinical recommendations.

## 1. Introduction

*Pseudomonas aeruginosa* (PA) is a pathogenic Gram-negative bacterium and leading cause of healthcare-associated infections (HAIs) around the world [1,2,3]. In the United States and Europe, PA accounts for 7.1% and 8.9% of all HAIs, respectively [4,5]. Meanwhile, some regions within Europe have reported even higher rates, with PA responsible for 10.5% of all HAIs in Spain [3]. PA is known to cause a variety of serious infections, including nosocomial pneumonia, bacteremia, urinary tract infections, and surgical site infections [1,2,3], with nosocomial pneumonia and bacteremia having mortality rates greater than 35% [1]. PA can form biofilms on catheters and tubes [1], putting patients with indwelling catheters and endotracheal tubes at increased risk for infection. Additionally, individuals with co-morbid conditions such as bronchiectasis, chronic obstructive pulmonary disease, immunocompromised status, and neutropenia have an increased risk of hospital-associated PA infection [1,2,3]. In particular, it is estimated 70% of adult cystic fibrosis patients are chronically colonized with PA [2].

In addition to its relatively high prevalence and potential for serious infection, multidrug-resistant (MDR) PA strains are particularly concerning for human health [6,7,8]. Clinically, MDR strains negatively affect patient outcomes and are associated with increased mortality and morbidity [1,2] and healthcare costs [9]. Since PA poses a significant threat to individual and population health around the world, it is important that clinicians treat PA infections to maximize patient outcomes and minimize the selection of resistant isolates.

Currently, a variety of antibiotic classes are used in the treatment of PA infections, including beta-lactams (BLs), fluoroquinolones (FQs), aminoglycosides (AGs), and, rarely, colistin, in combination therapy or alone in monotherapy [1]. However, given these choices, no firm standard treatment guideline exists. Empiric therapy with two antipseudomonal agents from different drug classes is recommended for critically ill patients with known or suspected PA bacteremia [10]. After microbiologic susceptibility testing is performed, it is recommended to de-escalate and initiate appropriate definitive therapy with the single agent that is most active against the infecting strain and has the least propensity to select resistance [10]. In practice, the combination of hospital-specific antibiogram data and patient characteristics—including allergies, comorbidities, and renal function—may direct clinicians to select the appropriate antimicrobial.

Much effort has focused on comparing the therapeutic efficacy of monotherapy versus combination therapy in the treatment of PA infection, yet the results are generally considered controversial [1,11]. On the other hand, few systematic reviews have compared the therapeutic effects of different antibiotic classes given as monotherapy for PA infection [12,13]. To our knowledge, none have evaluated BL versus FQ, the two most commonly used antipseudomonal classes. A systematic review and meta-analysis were performed to evaluate the association between mortality, bacteriological eradication, and the clinical success and treatment of PA infection with BL agents versus FQ agents as active, definitive monotherapy in adult inpatients. The goal was to provide clinically relevant information to help guide clinicians on which drug class to select in the definitive treatment of PA infection so that patient outcomes could be optimized.

## 2. Methods

This systematic review and meta-analyses were performed and reported in accordance with the Preferred Reporting Items for Systematic Review and Meta-Analysis Protocol guidelines [14]. This research was exempt from Institutional Review Board approval.

### 2.1. Selection Criteria and Definitions

As an overarching framework, the PICOT (population, intervention, comparison, outcome, time) model was implemented to define the inclusion and exclusion criteria for this systematic review and meta-analysis. Studies with populations of adult inpatients infected with PA were included. Studies with comparisons between definitive BL or FQ monotherapy, active against the infecting PA strain, were eligible for inclusion. Based on precedence in the literature, BL monotherapy was defined as BL ± beta-lactamase inhibitor [15]. Studies that reported outcomes of inpatient mortality, microbiological eradication, or clinical response were eligible for inclusion. No restrictions were placed on year of publication or when the study was performed. No restrictions were placed on geographical location. Case-control, cohort, and randomized controlled studies were all eligible for inclusion. Case reports and case series were excluded. Any studies that failed to address all the components of our PICOT model or had incomplete data were excluded. Only available, full-text published studies in English were eligible for inclusion.

### 2.2. Information Sources and Search Strategy

With the assistance of a research librarian, comprehensive literature searches of Medline and Scopus electronic databases were performed. Hand searches of the Cochrane Database of Systematic Reviews, PubMed, Google Scholar, and bibliographies of relevant articles and meta-analyses were also performed. The search strategy was formulated by three study investigators (CD, ER, and RW) and executed by a research librarian at Creighton University Health Sciences Library (CUHSL). A combination of key terms and MESH terms were formulated into the search, including beta-lactams (aztreonam, cefepime, ceftazidime, imipenem, meropenem, doripenem, piperacillin, tazobactam, OR piperacillin/tazobactam), fluoroquinolones (ciprofloxacin OR levofloxacin), monotherapy, *Pseudomonas aeruginosa*, *Pseudomonas* infections, *Pseudomonas*, nosocomial, and hospital acquired. A full, detailed record of the search strategy is included in the Appendix A.

### 2.3. Screening and Methodological Assessment

The titles and abstracts of the literature search results were screened for eligibility and annotated in Microsoft Word by one study investigator (ER) in accordance with the PICOT-based, predefined selection criteria. Eligible studies were selected, and full-text articles were retrieved via available electronic sources, CUHSL stacks, or interlibrary loan. Full-text versions of the selected studies were reviewed by one investigator (ER) for inclusion in the meta-analysis. The final population of studies included in the meta-analysis was reviewed and assessed for methodological quality and risk of bias independently by two reviewers (ER and RW) using the NIH National Heart, Lung, and Blood Institute Study Quality Assessment Tools for Controlled Intervention Studies or Observational Cohort and Cross-Sectional Studies for randomized control or cohort studies, respectively [16]. These assessment tools include an itemized list of fourteen questions with explanations of their significance but lack a grading algorithm. Reviewers attributed a quality rating of good, fair, or poor based on the number of NIH criteria the study met and the importance of those criteria per NIH guidance. When finished attributing study ratings, reviewers met to discuss the articles and reach consensus on any rating discrepancies. If consensus was not achieved, discrepancies were settled by third party arbitration (CD).

### 2.4. Data Collection and Outcomes Assessed

Unadjusted data were independently manually extracted by two study investigators (ER and RW) and compared in duplicate. Mortality was the primary outcome assessed for observational studies, whereas bacteriological eradication and clinical success were the primary outcomes assessed for controlled studies. These outcomes were selected because they are traditionally used in studies comparing antimicrobial efficacy and reflect important clinical variables and/or therapeutic efficacy. For studies in which causative organisms of infection other than PA were reported, only PA-specific data were extracted and used in the analysis. If an intent-to-treat (ITT) analysis was reported, ITT data were utilized in the meta-analysis. For mortality, 28-day and 30-day mortalities were included in the same data analysis. Based on consistency among definitions between studies, the definition of microbiological eradication included confirmed eradication and presumed eradication, and the definition of clinical success included clinical cure and improvement. We planned to stratify the data based on specific agents within a drug class and perform subgroup analyses if possible.

### 2.5. Statistical Analysis

All extracted data were entered manually into Review Manager 5.3. The decision to use a fixed- or random-effects meta-analysis was based on between-study heterogeneity as indexed by τ^2^ and I^2^. Effect sizes for outcomes from the retrospective cohort studies are reported as odds ratios (OR), whereas effect sizes from the randomized controlled trials are reported as risk differences (RD). All effect sizes are reported along with a 95% confidence interval (CI). Individual study and composite effects are displayed in separate forest plots for each outcome. Following a fixed-effects meta-analysis, publication bias was evaluated visually via funnel plots.

## 3. Results

### 3.1. Selected Studies

The literature search resulted in a total of 397 potentially relevant studies for screening, ranging in year of publication from 1981 to 2019 (Figure 1). Six additional relevant studies were identified via hand searching. Twenty-four articles were selected for full-text review, and ultimately, six studies were included in the systematic review and meta-analysis: three cohort [17,18,19] and three randomized control studies [20,21,22]. These six studies included 338 total patients. Common reasons for study exclusion included the lack of PA-specific data, no FQ or BL comparison or treatment arm, and incomplete data (Figure 1).

### 3.2. Methodological Quality of Studies

Upon evaluation of methodological quality, two studies were rated good, three rated fair, and one rated poor (Table 1). Of the cohort population, the study features we identified that increased the risk of bias included failure to report sample size justification and power descriptions, incompletely or inadequately defined outcome measures, and failure to adjust for potentially confounding variables. Of the randomized control population, study features identified to increase the risk of bias included designs allowing patients to receive non-protocol antimicrobial agents, options to receive vancomycin and/or metronidazole, non-ITT analysis, incomplete or no blinding, dropout rates higher that 20%, failure to report sample size justifications and power descriptions, and study details that were not reported or inadequately reported (see Appendix A).

### 3.3. Study Characteristics

The characteristics of the studies included in the meta-analysis are shown in Table 1. All studies evaluated hospitalized patients. Unlike the others, the study design of Siami et al. allowed for initially hospitalized patients to be discharged and later evaluated in an outpatient setting. All cohort studies selected patients with bacteremia, while the randomized control studies selected patients with pneumonia [20,21] or skin infection [22]. In all six studies, most patients were male, and a large proportion of patients were older than 50. Patients selected in the cohort studies shared similar characteristics including malignancy and immunosuppressed status at varying percentages between studies. Ciprofloxacin monotherapy constituted at least part of the FQ arm in all but one study [22]. A variety of BLs were used as monotherapy in the studies, including cephalosporins, carbapenems, and penicillins. All six studies incorporated the BL-beta-lactamase inhibitor drug when studying penicillins. Notably, all but two studies included cases of polymicrobial infections. Furthermore, Kuikka et al. excluded cases of polymicrobial bacteremia but reported almost one-fourth of the bacteremia patients had other infections. Although all three randomized control studies defined bacteriological eradication as eradication plus presumed eradication, the definitions of these two terms varied between the studies.

### 3.4. Mortality

A total of 211 patients with PA bacteremia from the three cohort studies were included into the meta-analysis comparing the effects of BL and FQ monotherapy on mortality. Among the six total treatment arms, mortality ranged from 0% to 32%. Four patients (9.8%) who received FQ monotherapy died compared to 50 (29.4%) who received BL monotherapy. The results of a fixed-effects meta-analysis indicate that FQ monotherapy resulted in significantly higher survival compared to BL monotherapy (OR, 3.65; 95% CI, 1.27–10.44; *p* = 0.02; Figure 2). Visual analysis of the associated funnel plot indicates no apparent publication bias (Figure 2).

### 3.5. Bacteriological Eradication

A total of 127 patients with PA pneumonia or skin infection from three randomized control studies were included in the meta-analysis comparing the effects of BL and FQ monotherapy on bacteriological eradication. Within six total treatment groups, bacteriological eradication ranged from 25% to 50%. A total of 28 patients (41.8%) who received FQ monotherapy were found to have culture-confirmed or presumed PA eradication compared to 21 (35.0%) who received BL monotherapy. The fixed-effects meta-analysis indicates that FQ monotherapy was not associated with increased bacteriological eradication compared to BL monotherapy (RD, 0.07; 95% CI, −0.09 to 0.24; *p* = 0.39; Figure 3), which was expected given that all of the studies had 95% CIs for their respective RD that included zero (a statistically similar result was observed when excluding Siami et al. [22] due to poor methodological quality; RD, 0.07; 95% CI, −0.12 to 0.26, *p* = 0.49). Based on the composite risk difference, the number needed to treat (NNT) was 15, indicating that 15 patients would need to be treated with FQ monotherapy for one patient to benefit from BL monotherapy. Visual analysis of the associated funnel plot indicates no apparent publication bias (Figure 3).

### 3.6. Clinical Success

Torres et al. was the only study to have PA-specific data comparing the effects of BL and FQ monotherapy on clinical response. Twenty-six patients with PA pneumonia were included in the report. Ten patients (71.4%) who received ciprofloxacin monotherapy were found to be clinically cured or improved compared to 8 (66.7%) who received imipenem-cilastatin (RD, 0.05; 95% CI, −0.31 to 0.40; *p* = 0.79; NNT = 20).

## 4. Discussion

A systematic review and meta-analysis of published studies was performed that compared the therapeutic efficacy of BL monotherapy with FQ monotherapy for PA infection in adult inpatients with the purpose of identifying apparent trends with respect to mortality, bacteriological eradication, and clinical success. The results demonstrate patients receiving FQ monotherapy had higher survival in PA bacteremia, but not higher rates of bacteriological eradication in pneumonia and skin and soft tissue infections. Based on one study’s data of 26 total patients, FQ monotherapy was indicated to have no benefit over BL monotherapy on clinical success rates in PA pneumonia [21].

Unadjusted data were used in the mortality analysis due to the lack of adjusted data accounting for potential confounding variables. Therefore, other inherent risk factors for mortality were not accounted for in the reported data. Wu et al. reported patient demographic data specifically for the BL and FQ arms. The BL arm had a greater percentage of patients with septic shock, immunosuppression, and higher mean Pitt bacteremia and APACHE II scores, whereas the FQ arm had a greater percentage of patients with malignancy [19]. Differences in Pitt bacteremia and APACHE II scores were statistically significant, meaning the BL group had more critically ill patients [19]. Unfortunately, treatment arm-specific patient demographics were not reported in the other two studies. In addition to the retrospective nature of the included studies, these discrepancies may have biased our results to an unknown extent.

For the meta-analysis on bacteriological eradication, all included studies reported the presence of polymicrobial infection, which often occurs in patients with ventilator-associated pneumonia [20]. This fact likely increased the external validity of the results. All three studies included an unknown percentage of patients who were allowed to receive non-protocol antimicrobials. Two of three studies allowed for the option for protocol-allowed vancomycin and/or metronidazole. These study designs may have introduced a certain level of risk for bias in the results. Although treatment arm-specific patient demographics were reported in all the studies, none of the studies further stratified these data for infection caused by PA specifically. Therefore, we could not determine if any confounding variable existed that would impart a potential advantage or disadvantage for bacteriological eradication.

It is worth noting the Clinical & Laboratory Standards Institute (CLSI) changed the minimum inhibitory concentration (MIC) breakpoints for FQ in 2019 [23]. Large-scale surveillance studies have demonstrated that ciprofloxacin resistance increased from 11.9% to 17.3% among *P. aeruginosa* strains between 1999 and 2008 [24]. Llanes et al. demonstrated that among *P. aeruginosa* strains overexpressing efflux pumps, 85.9% of those tested were susceptible to ciprofloxacin, using pre-2019 CLSI susceptibility breakpoints of ≤1 mg/mL. [25] Low-level FQ resistance can be undetectable with the previous breakpoints and can serve as a first step in the development of higher-level resistance. It is unclear how this change could have affected the results, as all the studies were published prior to the FQ breakpoint revision of 2019, but it is important to acknowledge the CLSI change.

Strengths of this meta-analysis include the use of NIH quality assessment tools and adherence to a well-defined PICOT model and the PRISMA reporting guidelines for systematic reviews and meta-analyses. Additionally, the variations in study year, geography, and patient demographics among the included studies enhance the external validity of this study. However, its limitations include a paucity of literature comparing the efficacy of BL and FQ as monotherapies, small sample sizes for each individual study, differing definitions of terms and outcomes among studies, incomplete data, the lack of gray literature search, and known and unknown discrepancies in patient demographics between treatment arms for infections caused specifically by PA. The latter is explained by the fact that not all studies were designed with the purpose of making direct comparisons between BL and FQ monotherapy or studying PA-specific infection (rather, some examined all causative pathogens of a given infection type). Additionally, the studies evaluated definitive therapy, and the efficacy of empiric therapy could have impacted the results. Consequently, these results should be interpreted with caution.

Overall, little research has focused specifically on comparing the therapeutic efficacy of BL and FQ drug classes as active, definitive monotherapy for PA infection. To our knowledge, this is the first meta-analysis to do so. The vast majority of systematic reviews compares combination therapy and monotherapy but results from existing studies are contradictory and controversial [1,11]. Although combination therapy is recommended in certain cases of PA infection for empirical therapy [10,11], de-escalation to a single active agent is encouraged, as this may decrease the potential for adverse events and antimicrobial-associated toxicity and reduce the development of resistance [11]. De-escalation to monotherapy is also consistent with antibiotic stewardship program objectives [24]. Regardless, more research comparing monotherapy in PA infection is needed.

When selecting a preferred drug class for PA infection, no overarching recommendations exist, and our results do not come close to bridging the gap. More likely, no one drug class or antimicrobial agent is the ideal choice. Antimicrobial resistance is in flux and varies by regions, so local antibiogram data are essential to selecting drug therapies. Once PA antimicrobial susceptibility testing is complete, the interplay between patient characteristics and drug features remains critically important in selecting definitive therapy. Furthermore, definitive antimicrobial therapy for PA must minimize the potential of selecting resistance.

Overall, these results provide insight into the therapeutic efficacy of BL and FQ drug classes as active, definitive monotherapy for PA infection in adult inpatients but fall short of offering definitive answers. The results suggest FQ monotherapy is associated with significantly higher survival rates compared to BL monotherapy, but more rigorous research is required to make definitive conclusions. Clinicians should continue to weigh the pros and cons of drug classes and individual agents for a particular patient when selecting definitive therapy for PA infection.

## Figures and Tables

**Figure 1 antibiotics-10-01483-f001:**
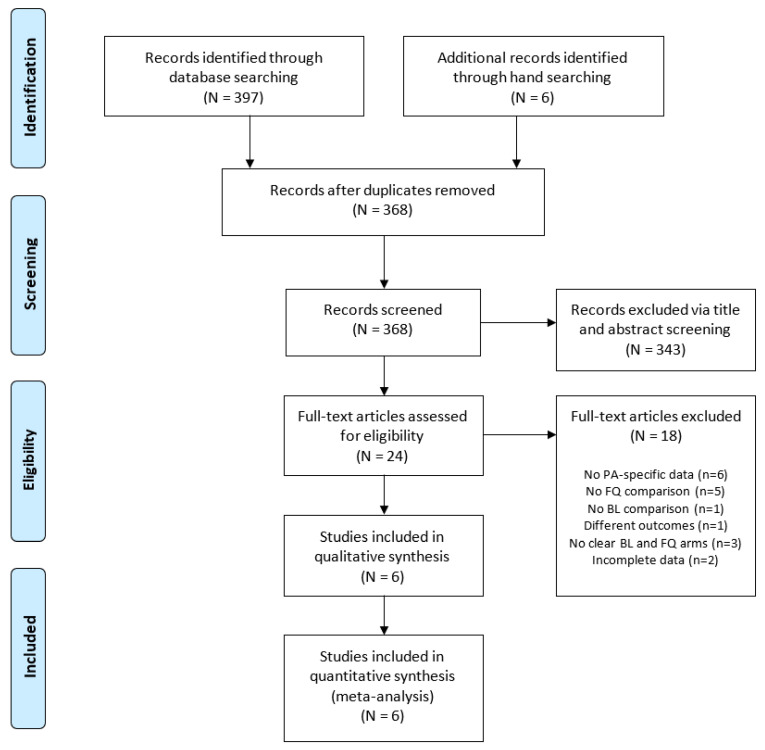
Flow diagram of study selection process.

**Figure 2 antibiotics-10-01483-f002:**
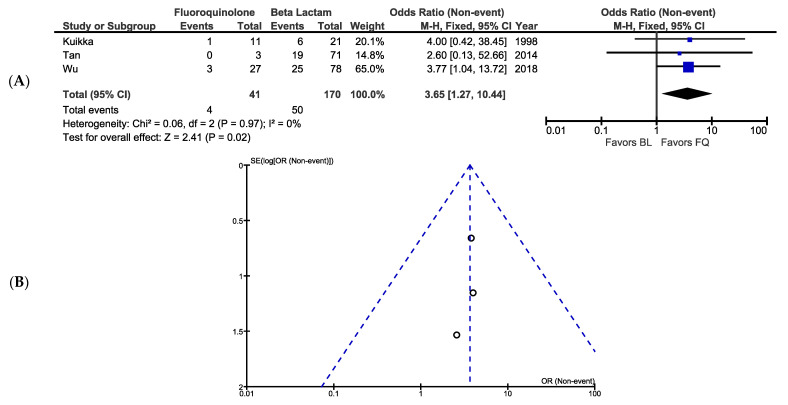
(**A**) Forest plot showing FQ monotherapy is associated with significantly improved survival compared to BL monotherapy using a fixed-effects model. (**B**) Funnel plot showing the studies included in the meta-analysis.

**Figure 3 antibiotics-10-01483-f003:**
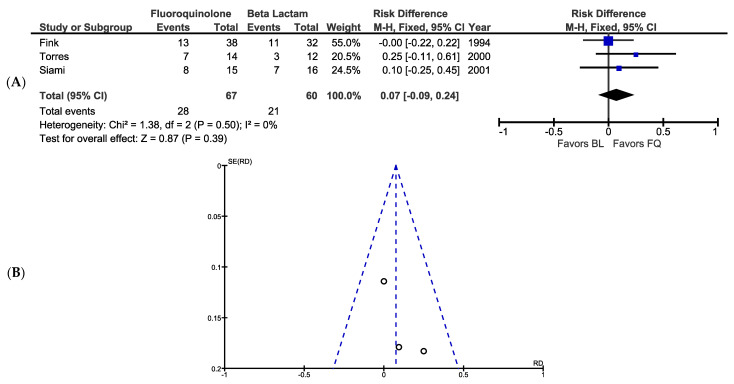
(**A**) Forest plot showing FQ monotherapy is associated with similar bacteriological eradication compared to BL monotherapy using a fixed-effects model. (**B**) Funnel plot showing the studies included in the meta-analysis.

**Table 1 antibiotics-10-01483-t001:** Characteristics of selected studies.

Study, Year	Study Design; Record Years	Study Location; Setting	Quality Rating	Infection Type; Mode of Acquisition	Outcomes	Patient Demographics ^b^	BL Arm: # of Patients; Drugs	FQ Arm: # of Patients; Drugs	Polymicrobial Infections	Pertinent Definitions ^⌿^
Kuikka et al., 1998 [17]	Cohort, retro; 1976–1982 & 1992–1996	Finland; Hospital (inpatients)	Fair	Bacteremia w/sepsis; nosocomial (90%) & community acquired	30 d mortality	63% male, 46% >60 y/o, 34% hematologic malignancy, 16% nonhematologic malignancy, 30% ICU; 37% systemic corticosteroid therapy; 35% cytotoxic therapy	21; carbenicillin, pipercillin (+tazobactam), ceftazidime, imipenem, meropenem	11; ciprofloxacin	N (for bacteremia), 24% other infection	Definitive therapy = 7 d or until death
Tan et al., 2014 [18]	Cohort, retro; 2007–2008	Singapore; Hospital (inpatients)	Fair	Bacteremia; nosocomial (45%), healthcare-associated (32%), community acquired (23%)	30 d mortality	59% male, 65 median age, 30 median SAPS II score, 1 median Pitt bacteremia score, 18% ICU, 44% active empirical therapy, 19% cancer, 9% HIV/AIDS	71; ceftazidime, piperacillin-tazobactam, carbapenems, piperacillin, aztreonam	3; ciprofloxacin	Y, 19% of patients receiving monotherapy	Definitive therapy = 2 d after culture results
Wu et al., 2018 [19]	Cohort, retro; 2013–2014	Taiwan; Hospital (inpatients)	Good	Bacteremia; nosocomial (66%), healthcare-associated (24%), community acquired (8%)	28 d mortality	71% male, 66 mean age, 64% malignancy, 21 mean APACHE II score, 43% septic shock, 3 mean Pitt bacteremia score; 30% chemotherapy, 17% steroid use, 16% neutopenia, 79% appropriate empirical therapy	78; piperacillin-tazobactam, ceftazidime, cefepime, imipenem-cilastatin, meropenem, doripenem	27; ciprofloxacin, levofloxacin IV or PO	N	Definitive therapy = >3 d & for >50% of treatment time
Fink et al., 1994 [20]	Randomized control, DB; 1990- 1992	USA; Hospital (inpatients)	Good	Severe pneumonia; nosocomial (78%) & community acquired	Bacteriological eradication	70% male, 59 mean age, 79% ICU, 17.6 mean APACHE II score, 15% bacteremia	32 *; imipenem-cilastatin ^v^	38 *; ciprofloxacin ^m,v^	Y, 50% of non-ITT population	Bacteriological eradication = eradication + presumed eradication
Torres et al., 2000 [21]	Randomized control, OL; NR	Spain; Hospital (inpatients)	Fair	Severe pneumonia; nosocomial	Bacteriological eradication, clinical response	74% male, 62 mean age, 13.8 mean APACHE II score	12; imipenem-cilastatin	14; ciprofloxacin	Y, 24% of microbiologically and clinically evaluable population	Bacteriological eradication = eradication + presumed eradication; Clinical success = cure +improvement
Siami et al., 2001 [22]	Randomized control, IB; NR	USA & Canada; Hospital (inpatients ^a^)	Poor	Severe SSTI (includes spontaneous, wound, and diabetic foot); NR	Bacteriological eradication	71% male, 53 median age, 41% spontaneous, 38% wound, 18% diabetic foot	16; piperacillin-tazobactam ^v^ w/PO option (amoxicillin-clavulanate) after 3 d	15; clinafloxacin w/PO option after 3 d	Y, 55%	Bacteriological eradication = eradication + presumed eradication

DB = double-blinded, OL = open label, IB = investigator-blinded, NR = not reported, SSTI = skin and soft tissue infection. ^a^ = patients were later discharged and evaluated in an outpatient setting; ^b^ = For Kuikka et al., represents population with PA bacteremia data (n = 132); for Tan et al., represents population with PA bacteremia receiving monotherapy data (n = 77); for Wu et al., represents population with PA bacteremia data (n = 105); for Fink et al., represents ITT population data (n = 402); for Torres et al., represents non-ITT, study population data (n = 75); for Siami et al., ITT population data (n = 409). For randomized control studies, data represent calculated averages of the two treatments arms, rounded down. * = ITT population, ^v^ = option for vancomycin, ^m^ = option for metronidazole. **^⌿^** = For randomized control studies, definitions of eradication and presumed eradication differ.

## Data Availability

The data is available from authors upon request.

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
