# Peer review of "Beta-Lactam vs. Fluoroquinolone Monotherapy for *Pseudomonas aeruginosa* Infection: A Systematic Review and Meta-Analysis"

_antibiotics, 2021, doi:10.3390/antibiotics10121483_

Round 1

Reviewer 1 Report

Here I present the review of the paper entitled “Beta-lactam vs Fluoroquinolone Monotherapy for Pseudomonas aeruginosa Infection: A Systematic Review and Meta-analysis” submitted to Antibiotics.

This is a review of monotherapy strategies for P. aeruginosa infection. Topic is novel and bring advancement to the field. Although interesting paper needs improvement

Specific issues:

  • Graphical abstract in present form does not serves its function. It is simply flow chart form PRISMA guidelines.
  • (line 70-71) “Meanwhile, some regions within Europe have reported even higher rates [3]”. Please specify what higher rates mean.
  • Has study been registered, for example in Prospero database?
  • Table 1, figure 2&3 are in poor resolution

Author Response

Thank you very much for your review. Please see the attachment for the authors' responses. 

  • Graphical abstract in present form does not serves its function. It is simply flow chart form PRISMA guidelines.
    • This was an error, and the graphical abstract is now included.
  • (Line 70-71) “Meanwhile, some regions within Europe have reported even higher rates [3]”. Please specify what higher rates mean.
    • PA is responsible for 10.5% of all HAIs in Spain. We have included this specification in the manuscript.
  • Has study been registered, for example in Prospero database?
    • We did not register with Prospero, but we did follow the Prisma statement to the letter (which, is based on Prospero elements).
  • Table 1, figure 2&3 are in poor resolution
    • We have improved the resolution of the tables and figures noted and inserted them into the revised manuscript as well as submitted them as attachments.

Reviewer 2 Report

Pseudomonas aeruginosa (PA) is a leading cause of fatal bacterial infection and there are two main classes of antibiotics mostly used, beta-lactams (BLs) and fluoroquinolones (FQs), active in combination or as monotherapy. The authors performed a systematic review and meta-analysis to indicate the therapeutic efficacy of BLs and FQs in eradicating PA infection. FQ monotherapy emerged to be more effective in determining improved patients'survival and bacterial eradication. The aim of this study is important, it is well conducted and clearly exhibited and I accepted it in the present form. 

Author Response

Thank you very much for your review. We have improved this manuscript according to other reviewers comments.

Reviewer 3 Report

Dear Editor 

Thank you for this opportunity in your esteemed journal 

The contents are interesting and fall well within the journal scope. However, the following issues must be well addressed.

  • Bacterial resistance from 1981 to 2019 was dramatically changed. Hence, this broad timeline seems inadequate
  • Also, PA infection is affected by several environmental conditions which direct the review process to focus on a specific population 
  • PA cause several diseases, hence the review would be improved upon concentrating on specific disease and the treatment regimen
  • Authors chose two antibiotic classes to compare between them, Why these two classes were specifically chosen

Author Response

Thank you all for your comments on our manuscript “Beta-lactam vs Fluoroquinolone Monotherapy for Pseudomonas aeruginosa Infection: A Systematic Review and Meta-analysis” and consideration for publication in Antibiotics. This letter will cover our responses to the reviewers’ comments and questions, addressed one by one.

  • Bacterial resistance from 1981 to 2019 was dramatically changed. Hence, this broad timeline seems inadequate.
    • The reviewer notes an important point.  The manuscript Discussion did note the history of breakpoints for PA to FQ particularly.  However, CLSI and NCCLS prior to that, did not lower the breakpoints for FQ against PA until 2018.  Additionally, the BL breakpoints are the same throughout the time-period.  Thus, the authors believe that this represents the state of antibiotic usage for PA infections. 
  • Also, PA infection is affected by several environmental conditions which direct the review process to focus on a specific population.
    • These environmental conditions would be similar in both treatment populations (BL compared to FQ) and thus the authors believe that this does not affect the results of the systemic review and meta-analysis. 
  • PA cause several diseases, hence the review would be improved upon concentrating on specific disease and the treatment regimen.
    • The aim of this project was to determine if there were any efficacy differences when patients infected with PA received BL compared to FQ therapy.  Our objective was to use systemic review and meta-analysis as the comparison between BL and FQ therapy.  Therefore, the authors reviewed all the literature involving efficacy of FQ compared to BL therapy for PA infections.  The authors did not want to limit the analysis based on a specific infection or treatment regimen as this would limit the analysis.
  • Authors chose two antibiotic classes to compare between them, Why these two classes were specifically chosen?
    • Beta-lactams and fluoroquinolones are the two most commonly used antimicrobial classes for treatment of PA infection. For this reason, it was decided to compare the two. Generally, aminoglycosides are avoided in monotherapy, and other agents are far rarer to use [Lynch et al, 2017].

Reviewer 4 Report

My specific comments are mentioned below

  1. Exclusion criteria is not clear.
  2. Why author has included a study with poor quality for meta-analysis?
  3. Is only 03 studies included as indicated by forest plot?
  4. Is 3 studies enough to make a valid conclusion?
  5. Funnel plot is not valid for 3 studies. At least 10 studies are required to get information from funnel plot regarding publication bias.
  6. Author should discuss already published meta-analysis on this or similar topic in discussion section.

Author Response

Thank you very much for your review. Please see the attachment for the authors' responses. 

  • Exclusion criteria is not clear.
    • To address this comment, we included a statement in the Methods section detailing how studies that failed to address our PICOT model or had incomplete data were excluded. Common reasons for exclusion are noted in Figure 1, but to highlight these reasons, we added an additional statement in the Results section addressing this concern.
  • Why author has included a study with poor quality for meta-analysis?
    • We included the Siami study in the meta-analysis to provide a complete picture of research conducted specific to bacteriological eradication. To address the reviewer’s comment, we have included in the Bacteriological Eradication section a comment on the pooled mean difference after omitting the Siami study. Briefly, the overall risk difference and inference did not change, but (as expected) the confidence interval is wider (risk difference: 0.07, 95% CI:-0.12 to 0.26, p = .49).
  • Is only 03 studies included as indicated by forest plot?
    • In all forest plots, we included all studies that measured the outcome of interest (e.g., survival, bacteriological eradication).
  • Is 3 studies enough to make a valid conclusion?
    • The reviewer has made a thoughtful comment. First, three studies for survival and bacteriological eradication are those that met inclusion criteria. In the Discussion, we had noted that a paucity of literature exists comparing the efficacy of BL and FQ as monotherapies. Further, throughout the Discussion, we chose our words carefully to not overstate the findings. That said, in the revision, in the Discussion, we have made this more clear when stating limitations.
  • Funnel plot is not valid for 3 studies. At least 10 studies are required to get information from funnel plot regarding publication bias.
    • To our knowledge, there is no minimum sample size requirement for a funnel plot because the pseudo-confidence interval that creates the funnel is a function of the (inverse) standard error on the y-axis. As such, the funnel plot is just an observe-and-report image that is not a function of small number of studies or statistical power. That said, it is difficult to obtain a meaningful appraisal of publication bias with a small number of studies, but as mentioned in response to the reviewer’s previous comments, we have reported on all studies that met inclusion/exclusion criteria.
  • Author should discuss already published meta-analysis on this or similar topic in discussion section.
    • There have been no previously published systematic reviews or meta-analyses on the comparison between different monotherapies for treatment of PA infection. This is the first systematic review to address this topic. We mention that the published studies that compare combination therapy versus monotherapy have contradictory results, but further discussion would be unrelated to our study’s focus.

Round 2

Reviewer 4 Report

Author has addressed my comments in the revised manuscript